# Role for Selenium in Metabolic Homeostasis and Human Reproduction

**DOI:** 10.3390/nu13093256

**Published:** 2021-09-18

**Authors:** Albaraa Mojadadi, Alice Au, Wed Salah, Paul Witting, Gulfam Ahmad

**Affiliations:** 1Molecular Biomedicine, Charles Perkins Centre, School of Medical Sciences, The University of Sydney, Sydney, NSW 2006, Australia; albaraa.mojadadi@sydney.edu.au (A.M.); alau4990@uni.sydney.edu.au (A.A.); wed.salah@sydney.edu.au (W.S.); paul.witting@sydney.edu.au (P.W.); 2Department of Anatomy, College of Medicine, King AbdulAziz University, Rabigh 21589, Saudi Arabia; 3Department of Anatomy, College of Medicine, Jeddah University, Jeddah 21959, Saudi Arabia

**Keywords:** selenium, thyroid, hormones, selenoproteins, body metabolism, obesity, fertility

## Abstract

Selenium (Se) is a micronutrient essential for life. Dietary intake of Se within the physiological range is critical for human health and reproductive functions. Selenium levels outside the recommended range have been implicated in infertility and variety of other human diseases. However, presently it is not clear how different dietary Se sources are processed in our bodies, and in which form or how much dietary Se is optimum to maintain metabolic homeostasis and boost reproductive health. This uncertainty leads to imprecision in published dietary guidelines and advice for human daily intake of Se and in some cases generating controversies and even adverse outcomes including mortality. The chief aim for this review is to describe the sources of organic and inorganic Se, the metabolic pathways of selenoproteins synthesis, and the critical role of selenprotenis in the thyroid gland homeostasis and reproductive/fertility functions. Controversies on the use of Se in clinical practice and future directions to address these challenges are also described and discussed herein.

## 1. Introduction

### 1.1. Elemental Selenium

Selenium (Se) is a metalloid (semi-metallic) trace element represented with the symbol Se and atomic number 34 in the periodic table [1,2]. Selenium was first discovered by Jöns Jakob Berzelius in 1817. Initially, it was regarded as a toxic element, but in 1957 Klaus Schwarz found that it is a beneficial and an essential trace nutrient [3]. Since the discovery of the link between Se consumption and resistance to diseases in grazing animals and human beings, it has undergone an extraordinary turnaround [4,5]. As it had been regarded toxic for such a long time, there has been considerable controversy regarding the harm and benefits of its consumption. Interestingly, Se shares similar physio-chemical properties to sulfur (S), as they are both classed as Group 16 (chalcogens) in the periodic table, where both elements have similar bonding strength, ionizing and redox potentials [6]. In addition, Se can couple with various elements (hydrogen, bromine, chlorine, fluorine, and phosphorus) and yields products that are analogous to the corresponding S-containing compounds in terms of both physical and chemical properties [7,8,9].

### 1.2. Selenium and Selenoproteins

Elemental Se is not biologically active per se. However, this trace element exerts its biological function(s) through its incorporation as selenocysteine and selenomethionine with subsequent integration into 25 selenoproteins encoded by the human genome [10,11] and 24 known proteins in rodents [12]. Thus, selenoproteins are Se-dependent proteins with a selenocysteine (Sec) amino acid residue at their active sites. Of the known selenoproteins most are involved/associated with redox homeostasis as well as redox regulation of transcription factors and signaling cascades linked to antioxidant response elements. However, around 50% of the selenoproteome are yet to be attributed to any known biological activity [13,14] and their role in biology remains to be clarified.

### 1.3. Significance of Different Forms of Selenium and Selenoproteins to Health

The trace nutrient is generally bioavailable in two forms: organic and inorganic Se. It is reported that the supplemented form of Se has an important influence on whether it is beneficial or harmful to organisms’ development and overall well-being [15,16,17]. Additionally, investigators have asserted that the form of Se and dosage have equal importance in defining the role for Se in physiological homeostasis. Nevertheless, it has been stressed that the form of Se used for consumption could be more important than its dosage [18].

While the main source of Se is consumption of Se-rich food, reports have indicated that there is a higher level of bioavailability in organic forms of Se (selenomethionine [SeMet] or selenocysteine [SeCys]) in comparison with inorganic forms (selenate [Me_2_SeO_4_] and selenite [Me_2_SeO_3_]) [19]. While organic types of Se have a 90–95% bioavailability, the bioavailability of inorganic Se was slightly lower, determined at 80–85% [20].

Despite the lower levels of bioavailability of inorganic types of Se, this form of the micronutrient may still be useful for synthesizing selenoproteins under stressful conditions or pathogenic insult [21,22]. Interestingly, organic Se has lower toxicity levels compared to inorganic selenite or selenate forms, which indicates that higher levels of bioavailability is associated with lower levels of toxicity [23], which may be pertinent when developing strategies to develop new synthetic bioavailable Se supplements.

### 1.4. Sources of Selenium

Selenium is a ubiquitous element in nature which can be found in all layers of the Earth as well as the atmospheric air generated mostly from volcanic gases [24]. Air in the environment is continuously and further enriched with Se due to several biological processes including biomethylation of Se by microbes and degradation of biological matter abundant in this material, yielding a variety of unstable Se compounds including dimethylselenium (DMSe), hydrogen selenide (H_2_Se), and selenium oxide (SeO_2_). The estimated Se concentration in the agricultural soil ranges from 0.33 to 2 mg/kg globally [25]. Soils generated from source rocks abundant in Se, such as limestone and sandstones, have been documented to have a relatively high Se content [26,27] and consequently agricultural land in regions rich in these rocks have higher Se content than other regional areas.

In mineral soil, irrespective of its thickness, the amount of Se approximates to 14 mg/kg [28]. A minute level of Se is found in groundwater, usually in the form of selenates and selenites, yet its concentration can substantially increase in seawater [27,29]. The higher concentration of Se in seawater is mainly due to Se filtration from parent rocks and run-off from intense soil fertilization with mixtures high in Se compounds [30]. Nonetheless, per the guidelines of the World Health Organisation [31], the safe level of Se in drinkable water is less than 10 μg/L [31].

The amount of Se contained within food sources can vary according to a range of factors including the soil and growing conditions in which the crops used to produce bread and cereals are grown, the fodder that animals consume, and the processing of these foods for human consumption. The primary sources of Se are bread, cereals, eggs, meat, fish, dairy products, fruits, and vegetables. Each of these sources will be elaborated below.

#### 1.4.1. Bread and Cereals

The amount of Se present within cereals and bread ranges from ~0.01–30 mg/kg [18]. The form of Se most common in wheat and bread is selenomethionine (detected in the range ~55–85% of total Se content), followed by selenocysteine (~4–12%) and selenate/selenite (~12–19%) [32,33]. The variation in the Se content in grains such as wheat is primarily dependent on the soil in which the crop is grown. For example, bread in New Zealand is made with Australian wheat which contains a good level of Se, because although wheat can be grown in New Zealand, it lacks Se due to the lack of Se in the soil in which it is grown.

#### 1.4.2. Meat, Fish and Eggs

Meat is typically a source of a relatively high amount of Se in omnivores, yet the amount of Se found in meat varies according to a range of different factors. Offal (animal organs) is particularly high in Se, especially sources that are derived from the liver and kidneys. According to a study by Juniper at el. (2008), the Se concentrations found within the kidney, liver, and heart tissue of bovine are 4.5, 0.93, and 0.55 mg/kg, respectively, while muscles of cattle contain 0.2 mg/kg of Se. Furthermore, the amount of Se in each meat product can vary according to the food consumed by the animal. For example, if cattle are supplemented with Se-enriched yeast in feed, the Se concentration in the muscle can increase up to ~0.6 mg/kg [34]. Selenomethionine (~50–60% of total extractable Se species) and selenocysteine (20–31% and ~50% of total extractable Se species in chicken and lamb, respectively) represent the primary forms of Se present in edible meat sources [1].

Selenium content in fish ranges from 0.1–5.0 mg/kg [18,35,36]. However, some species of marine life contain higher levels of Se than others. For instance, cod, shark, and canned tuna contain ~1.5, 2.0, and 5.6 mg/kg, respectively [35,36]. Selenomethionine (29–70%) and selenite/selenate (12–45%) are the types of Se commonly detected in fish [18,35,37]. As for poultry, each hen egg contains ~3–25 μg/kg Se [38]. In some cases, supplementary foods can help to increase Se content in eggs to up to ~0.34–0.58 mg/kg [39]. The Se species which are most frequently found in eggs are selenocysteine, selenomethionine, and sometimes selenite, with selenomethionine and selenocysteine representing the prevalent species (>50%) in egg whites and egg yolks, respectively [38].

#### 1.4.3. Milk, Dairy Products and Beverages

The amount of Se in meat and dairy products can vary significantly. In the United Kingdom, dairy products typically contain around ~0.01–0.03 mg/kg Se. The prevalent forms of selenoprotein found in bovine milk products are selenocysteine and selenite [40]. When the diet of cows is supplemented with Se-enriched yeast, the selenoprotein profile in the milk is altered, with selenocysteine, selenomethionine, and selenite representing the major forms of selenoprotein post-supplementation [40].

#### 1.4.4. Fruit and Vegetables

By comparison to protein-based sources, fruits and vegetables contain relatively low levels of Se. While onions contain selenite, garlic contains selenomethionine (53%), g-glutamyl-Se-methylselenocysteine (31%), Se-methylselenocysteine (12%) and selenate (4%) making a total Se content of only <0.5 mg/kg [41]. However, garlic and onions, when cultivated with a Se-rich soil exhibited enhanced Se accumulation ranging from 0.5 to 140–300 mg/kg. To summarize, Se content in vegetables, such as onions and garlic varies according to the level of Se enrichment during the cultivation process and the type of vegetable.

### 1.5. Nutritional Requirements of Selenium

Dietary reference intakes (DRI) and tolerable upper intake levels (UL) of Se vary across different countries of the world. For example, in the UK, the recommended intake is 75 µg/day for males and 60 µg/day for females. This is comparable with the 2016 recommendations of institutions based in Australia and New Zealand (www.nrv.gov.au/nutrients/selenium, accessed on 13 April 2021). However, according to the European Food Safety Authority (EFSA), the recommended intake of Se in the European Union (EU) is lower, 55 µg/day (UL 300), which is consistent with the Food and Nutrition Board (FNB) [42] in the USA (UL 400). By contrast, the World Health Organisation [31] recommends no more than 26 μg/day for females and 34 μg/day for males, which are considerably lower than those recommended by other organizations mentioned previously. Therefore, it is of critical importance to establish consistent information on Se daily requirements vital for normal body function.

Selenium toxicity, also known as selenosis can occur if Se intake exceeds the 400 μg/day threshold [43]. At even higher Se concentrations, Se can become pro-oxidant, leading to oxidative stress and damage to cellular components. Hence, maintaining a physiological concentration of Se within an optimum range is crucial to ensuring normal biological functions, while avoiding the deleterious effects from excessive Se intake.

## 2. Synthesis of Selenoproteins

### 2.1. Selenocysteine–tRNA [Ser]Sec Aminoacylation

Selenoprotein biosynthesis involves a two-step process including an initial formation of selenophosphate from HSe^−^, followed by incorporation into the translating protein chain during selenoprotein synthesis (refer to schematic Figure 1) [44]. First, selenocysteine is synthesised through serine misacylation of transfer RNA selenocysteine (tRNASec), followed by phosphorylation by phosphoseryl-tRNASer Sec kinase (PSTK) and dephosphorylation by selenocysteine synthase [45]. Next, the precursor undergoes a unique process termed translational recoding. Here the stop codon UGA in the mRNA sequence is decoded in the 3′ untranslated region to allow the insertion of selenocysteine (Sec) [46]. This highly regulated process is dependent on a downstream stem-loop-structure termed the selenocysteine insertion sequence (SECIS), which is recognized by SECIS-binding protein (SECISBP2) [44].

Binding of SECISBP2 to SECIS recruits selenocysteine-specific elongation factor (EFSec), which in turn recruits selenocysteine tRNA, facilitating selenocysteine to be incorporated into the growing polypeptide. Thus, selenoprotein synthesis is dependent on Selenocysteine–tRNA[Ser]Sec aminoacylation. A critical RNA element of selenocysteine injection machinery is Sec-tRNA Sec, which is the only tRNA that alone regulates expression within the selenoproteomes [47,48]. Sec-tRNA Sec is unique in terms of its length, aminoacylation, transcription, modification, and ribosome delivery.

Aminoacylation of Sec-tRNA is especially unusual in that it involves four enzymes rather than one for traditional tRNAs as shown in Figure 1a. Therefore, Sec-tRNA [Ser]Sec levels are heavily reliant on the bioavailability of Se within the cell. When Se levels are diminished, selenide is dedicated to creating fully mature selenocysteine-tRNA Sec. Under these conditions, Se-bioavailability would be limited for the synthesis of selenoproteins. Recently, another regulatory mechanism in low Se status was described [49,50,51]. This pathway involves the substitution of Cys for Sec in selenoproteins as SEPHS2 is also capable of generating thiophosphate rather than selenophosphate indicating some redundancy in the pathway and a potential switch in substrate (Se to S) under limiting Se bioavailability.

**Figure 1 nutrients-13-03256-f001:**
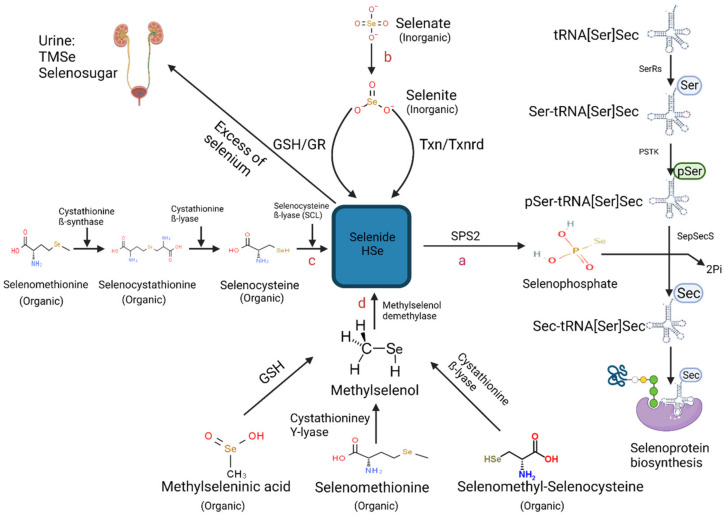
Schematic representation of the known metabolic pathways contributing to selenoprotein biosynthesis [52] (Created with BioRender.com (2021)). Pathways include (**a**) Aminoacylation of Selenocysteine–tRNA[Ser]Sec, (**b**) Chemical reduction of selenite into selenide by either the thioredoxin/thioredoxin reductase (TXN/TXNRD) system or of the glutathione peroxidase redox (GSH/GR) system (**c**) Catalyzation of selenomethionine-derived selenocysteine by three enzymes: cystathionine ß-synthase, cystathionine ß-lyase and selenocysteine ß-lyase (SCL) (**d**) Demethylation reaction of methylselenol (CH_3_Se) catalyzed by methylselenol demethylase. Abbreviations includes: (SPS2) selenophosphate 2 synthetase 2, (SerRS) seryl-tRNA synthetase, (PSKT) phosphoseryl-tRNA kinase, (SepSecS) Sec synthase, (TMSe) trimethylselenonium ion.

### 2.2. Hydrogen Selenide—The Core Precursor of Selenoprotein Synthesis

Selenoprotein synthesis is achieved through the absorption and catabolysis of Se-rich food, where hydrogen selenide (HSe^−^) is central to both inorganic and organic Se crucial to the biosynthetic pathway. A screen of the available data indicates three metabolic pathways have indeed been documented for the generation of HSe^−^, which include the substrates selenocysteine, selenite, and methylselenol derived from Se-rich food (Figure 1).

## 3. Selenium and Thyroid Function

### 3.1. Optimal Selenium Maintains Normal Thyroid Function

Selenium is a micronutrient essential to maintaining several cellular processes including the most basic biological process—DNA synthesis [53]. The thyroid has the highest Se concentration among all other endocrine organs, indicating the significance of the functions of thyroid [46,54]. In humans, optimal Se status is essential for the maintenance of thyroid health, metabolism of thyroid hormones (TH), and protection from thyroid disorders [10]. Though debated, numerous clinical studies have illustrated that Se supplementation had anti-inflammatory effects in patients with autoimmune thyroiditis, characterised by reduced anti-thyroid peroxidase complement-fixing autoantibody (TPOAb) levels and restored thyroid features [55,56]. Selenium is also well-known for incorporating into endogenous antioxidant systems, thereby acting as part of the homeostatic response to oxidative stress. However, Se toxicity occurs when intakes exceeds daily threshold that leads to redox imbalance in the cells as described above [43]. Hence, maintaining a physiological concentration of Se within an optimum range is crucial to ensuring normal thyroid function and subsequent production of key regulators important to metabolism.

### 3.2. Selenium-Containing Selenoproteins Mediate Thyroid Function

Selenium exerts its biological functions through its incorporation as selenocysteine into selenoproteins in humans [10,11] and rodents [12]. Selenoproteins have various biological roles in the thyroid, ranging from catalyzing enzymatic redox reactions, regulating thyroid hormone metabolism, and protecting against oxidative DNA damage induced by hydrogen peroxide (H_2_O_2_) and lipid hydroperoxide, and inflammation (Table 1) [57]. The significance of selenoproteins to thyroid health is reflected by the increased risk and mortality of thyroid-associated diseases in single nucleotide polymorphisms in selenoprotein genes [54]. For example, polymorphisms of glutathione peroxidase (GPX3) selenoprotein, which will be discussed later are associated with differentiated thyroid cancer [46,54].

### 3.3. Interplay between Selenoproteins and Thyroid Hormones Synthesis

Selenoproteins that contain enzymatic activities and are closely linked to thyroid derived metabolism are glutathione peroxidase (GPX), iodothyronine deiodinase (DIO), and thioredoxin reductase (TXNRD) [10], which are described in detail below.

**GPXs:** The GPXs are antioxidant enzymes with several biological roles, including hydrogen peroxide (H_2_O_2_) signaling, catalysis of hydroperoxide degradation, and maintenance of cellular redox homeostasis [57]. In the thyroid, GPX1, GPX3 and GPX4 are present and are primarily responsible for glandular protection against oxidative stress by converting excess oxygen free radicals (H_2_O_2_) involved in thyroid hormone synthesis into water (Figure 2) [43].

**DIOs:** Three DIO proteins are documented (DIO1, DIO2, and DIO3) within the thyroid and all three play a role in thyroid hormone metabolism, where TH has downstream metabolic regulatory roles including growth, differentiation, thermogenesis, and metabolism that are mediated by nuclear receptors of thyroid hormones expressed in various tissues [73]. Both DIO1 and DIO2 are responsible for the conversion of active TH, triiodothyronine (T3) from its inactive precursor, thyroxine (T4), by deiodination [10]. However, T3 produced by intracellular DIO2 activates T3-dependent gene transcription more effectively than that produced by DIO1. Unlike DIO1 and DIO2, DIO3 inactivates thyroid hormones through the conversion of T4 to reverse T3 (rT3). The regulation of DIOs expression is a feedback loop that involves T3 and thyroid-stimulating hormone (TSH), in which nuclear accumulation of T3 induces DIO1 gene expression at the transcription level [53].

**TXNRDs:** The TXNRDs selenoenzymes have a wide range of disulfide-containing substrates other than thioredoxin (Trx) and are responsible for the regulation of redox signalling system, transcription, immunomodulation, and cell growth factors [74]. In the Trx/TXNRD system, TXNRD reduce the active site disulfide of Trx (Trx-S_2_)—a high-capacity biological reductant for thioredoxin, Trx-(SH)_2_ using NADPH as the hydrogen donor [75]. The oxidoreductase activity of Trx-(SH)_2_ then reduces other protein sulfides to regenerate the original Trx-S_2_, which may activate DNA synthesis, regulate transcription factors of numerous inflammatory genes or downstream signaling for apoptosis in mammals [53,74]. The Trx system catalyzes the reduction of peroxiredoxin (Prx-S_2_) to form reduced peroxiredoxin, Prx-(SH)_2_, which then catalyzes the chemical reduction and degradation of H_2_O_2_ and organic lipid hydroperoxides thereby playing a role in host defense against oxidative and inflammatory stress and damage, and their associated disease processes [76].

As an example, Trx’s role as the electron donor in the formation of reduced Prx-(SH)_2_ highlights its importance as an antioxidant that indirectly protects against oxidative stress and associated disease processes including diabetes and obesity-related metabolic disorders [77]. In the thyroid, the Trx system was found to play a critical role in ameliorating the oxidative stress induced by excess iodide and restoring thyroid hormone synthesis, thereby underscoring its function in redox control and reestablishment of thyroid homeostasis [78]. Recently, lower expression of TXNRD1 along with the GPX1 antioxidant selenoprotein, and increased free radical production has been detected in thyroid cancer, compared to healthy thyroid tissue [79]. These results suggest that an insufficient or lack of antioxidant defense and consequently increased oxidative stress that arises from the failure of Trx/TXNRD system to maintain a balanced oxidant/antioxidant system in thyroid tissues, and that an adequate antioxidant defense is required for the maintenance of healthy thyroid tissue.

### 3.4. Preferential Synthesis of Selenoprotein in the Thyroid

Interestingly, there is a preferential synthesis of a specific selenoprotein type over the other during insufficiency of Se supply [80]. For example, DIO1 synthesis is favored to afford the efficient production of active T3 hormones, whilst other selenoproteins including GPX1 are only synthesized in case of high Se supply [81]. Moreover, in the case of Se deprivation, the distribution of Se to the thyroid, an endocrine organ, is preferentially favored, whereas Se pools in serum, liver, and muscles are rapidly lost [10,80]. Despite the thyroid’s capacity to retain and utilize Se efficiently, chronic Se deficiency adversely impacts the synthesis of selenoproteins and consequently the metabolism of thyroid hormone [80] that may lead to thyroid disorders.

### 3.5. Thyroid and Metabolic Homeostasis

As described earlier, thyroid hormone is widely known to regulate a range of metabolic processes throughout the body including growth, development, and notably, thermogenesis, basal metabolic rate, energy expenditure, and body weight [71]. While hyperthyroidism (excessive thyroid hormone levels) promotes a hypermetabolic state that leads to increased thermogenesis and basal metabolic rate that decreases body weight, hypothyroidism (reduced thyroid hormone levels) has opposing effects that ultimately causes an increase in body weight [71,82]. Therefore, a link between Se-thyroid and body weight can be established. For example, several observational studies have found an association between Se deficiency and obesity (defined by BMI and body fat percentage) due to impaired selenoprotein synthesis, metabolism, and hence thyroid hormone production; however, the underlying mechanisms are not completely understood [83,84,85,86].

More recent studies have even linked Se deficiency with other comorbidities of obesity including hypertension and impaired glycemic control [87,88]. Importantly, Se nanoparticles were recently found to alleviate hyperlipidemia and vascular injury in ApoE^-^deficient mice by regulating cholesterol metabolism and reducing oxidative stress [89]. These observations have prompted investigations in refining Se intake to optimize selenoproteins and thyroid hormone production in the thyroid and hence subsequent fat metabolism to achieve a healthy body composition and reduce comorbidities in patients with metabolic disorders [90]. Despite appropriate dietary Se intake in obese women, the blood Se levels were significantly lower than that in healthy individuals [91]. This can be due to increased oxidative stress in obese women requiring Se cofactors as part of antioxidants to protect cells against oxidative damage [92]. Notably, obesity is also characterized by chronic low-grade inflammation that inhibits liver’s production of selenoprotein P involved in transporting Se in plasma [91]. Therefore, optimal Se intake in obese patients was speculated to be higher (82.4–200 μg) than that in the general population but it remained difficult to estimate due to the limited availability of data [93].

As indicated above, the daily recommended Se intake in the general population varies due to geographical and racial characteristics, and dietary habits, thus a clear threshold for daily Se intake in obese population is undefined despite of the fact that Se is essential for thyroid functions, in modulating body’s response to inflammation and oxidative stress [90]. Nonetheless, increasing Se intake in populations with Se-deficient soil is of critical importance as impaired selenoprotein synthesis and thus expression and activity, induced by Se deficiency may contribute to adipocyte dysfunction and dysregulated lipid accumulation and lipolysis that ultimately leads to obesity and the development of comorbidities including diabetes and cardiovascular diseases [94]. High-fat diet can stimulate the activity of a selenoprotein DIO1 in white adipose tissue involved in the production of T3 from T4, whilst caloric restriction decreased the activity of DIO1 in white adipose tissue, which again emphasizes the critical role of DIO1 in regulating adipose tissue metabolism and accumulation [95].

### 3.6. Controversies around the Application of Selenium Supplementation in Ameliorating Obesity

The previous section has highlighted the potential of Se supplementation to complement with other weight loss strategies including hypocaloric diets and exercise to maximize weight loss and enhance the general health of obese populations. However, the effects of Se diets may vary according to the different forms and doses of Se resulting in differences in absorption and metabolic pathways, along with the length of supplementation and Se status of subjects. A recent study revealed that obese individuals, when they adopted a slightly hypocaloric diet for 3 months, showed reduced body weight and fat mass, however the weight loss and changes in body composition were more evident when coupled with selenomethionine supplementation [90]. Yet, another recent study showed conflicting results, where Se-rich food caused decrease in serum T3 level and compensatory increase in TSH through modulation of selenoproteins involved in thyroid hormone metabolism, and consequently the observed body weight gain [82].

These contradicting results could partly be explained by the fact that the increase in T3 levels and subsequent fat metabolism and weight loss would only manifest in Se-deficient individuals, which was consistent with the results from the first study that was based in Italy with Se-deficient soil and lower Se status. It was postulated that the high Se intake in populations with sufficient Se intake at the baseline could depress the activity of DIO1 involved in the production of active T3 hormone, leading to reduced T3 levels and thus body metabolism [96]. This theory was recently corroborated in an Indian observational study that revealed high serum Se, induced by high Se environment was associated with low DIO activity and serum T3 levels [97]. Despite the use of different study designs and methodologies, a majority of available data show that Se supplementation has no impact on T3 and TSH levels and body weight [82,98,99]. Therefore, future studies are needed to determine the potential of Se supplementation and a standardized chemical form and dose of Se that is most effective in enhancing body fat metabolism in obese populations, with follow-up studies to assess Se efficacy and pharmacological safety in the long term.

### 3.7. Thyroid Disorders Associated with Selenium Deficiency

Moderate Se deficiency has been linked to decreased thyroid function and increased occurrences of thyroid disorder as summarized in Figure 3 [46,61,100]. This is because a lack of Se causes a reduction of both DIO and GPX enzymatic activities. DIO, which is responsible for the conversion of T4 to its active form T3 becomes less active, resulting in decreased active TH synthesis. The lack of circulatory TH stimulates the hypothalamic–pituitary axis through positive feedback to release the thyroid stimulating hormone (TSH) [43]. Endogenous TSH then stimulates DIO’s activity to convert inactive T4 into more T3 hormone—a process that requires adequate H_2_O_2_ production to activate its oxidative properties [101]. However, under conditions where the accumulated H_2_O_2_ is not adequately removed by the less active GPX, due to Se deficiency, significant host damage occurs to thyrocytes followed by necrosis and fibrosis of the thyroid [43]. Even though the impacts of Se deficiency on general health can remain unimpaired, there are changes on thyroid metabolism occurring at the microscopic level [102]. For example, DIO1 gene deletion in mice results in aberrant circulating T3 and urinary and fecal excretion patterns of thyroid hormone metabolites including iodothyronines, despite the normal growth and development of these mice [53].

### 3.8. Selenium Supplementation and Thyroid Disorders

A growing interest in the beneficial role of Se in human health is now focused on disease-prevention. Despite the significant role of dietary iodine in normal thyroid functioning, patients with thyroiditis and severe iodine deficiency failed to respond to iodine supplementation treatment [103]. However, iodine supplementation, when used in combination with Se supplementation successfully reduced inflammatory markers in patients with autoimmune thyroiditis. These clinical outcomes suggest that Se deficiency is one of the major contributing factors in autoimmune thyroid disorders and further studies investigating the optimal Se type and level for supplementation are warranted.

In addition, low Se status is associated to an increased risk of autoimmune thyroiditis, Grave’s disease and goiter (enlarged thyroid gland) [46]. It is now documented that Se supplementation can clinically benefit patients with autoimmune thyroiditis and Grave’s orbitopathy [104]. For example, patients with chronic thyroiditis and mild Se deficiency taking 200 μg sodium selenite per day had a significant (40%) reduction in thyroid peroxidase antibodies (TPOAb) levels, when compared to the placebo group who had a 10% increase in TPOAb level [56]. A 6-month follow-up study revealed that patients who discontinued Se supplementation showed an increase in the TPOAb levels [105] suggesting that patients benefited from Se supplementation. Unfortunately, there is no consistency and consensus which form of Se and how much Se supplementation is optimal for thyroid and associated body functions. Nonetheless, an imbalance in the physiological levels of Se may lead to adverse outcomes (Figure 3). Owing to the strong linkage between thyroid hormones and reproductive functions, the next sections of this review will focus on the role of Se in human reproduction.

## 4. Selenium-Thyroid and Female Reproduction

The optimum reproductive ability relies on several variables, including genes, external ecological criteria, and individual diet [106]. Out of those, micronutrients diet is essential for numerous biological functions, including growth and reproductive capacity [107]. Moreover, small differences in micronutrient levels can dramatically affect essential physiological processes including fertility [106,107] Some studies have shown the connection between Se status and reproductive function in males [106,108] and females [106,109].

Currently there is scarce information on the significance of Se to female fertility. In female reproductive health, the production of a healthy oocyte involves a number of sequential steps. One primary step is folliculogenesis which is the developmental process of primordial ovarian follicles at birth into mature follicles after puberty. An essential step of folliculogenesis is the proliferation of granulosa cells, where Se has been shown to control the development of granulosa cells and the biosynthesis of one of the major female sex hormones, 17-estradiol (E2) in adult ovaries in vitro [110]. Recent research has also revealed that Se levels are higher in large healthy follicles, suggesting that they can play an important antioxidant role during follicular growth and proliferation [111]. Yet, the governing function that Se is thought to play in the purpose and growth of ovaries in a fetus is still obscure [112].

Thyroid hormones are essential for the optimization of mammalian reproduction and growth [113]. Inadequate synthesis of thyroid hormones has been associated with diminished fertility in humans and rodents, estrus cycle disruption, dysfunctional implantation and uterine defective architecture and other problems related to pregnancy health [113]. For example, higher DIO3 mRNA expression was reported in the uterus of pregnant rats [114]. An interesting outcome is that DIO3 was precise in expression time and location, i.e., intensely expressed in the uterus mesometrial and anti-mesometrial decidua on the 9th day of gestation indicating its role in implantation [114]. It has been speculated that the significantly high level of expression of DIO3 at implantation site protects fetal development from over-exposure to maternal thyroid hormones which can impact negatively on in utero development of the fetus [16]. Similarly, DIO3 activity also increases with advanced gestational age in human placental cells [16].

This available data clearly demonstrates the important role of uterine and placental tissues in normal pregnancy and the control of fetal exposure to maternal thyroid hormones [114]. Huang and his colleagues reported an elevated level of expression of DIO3 in different locations, i.e., syncytiotrophoblast and cytotrophoblast, placental and amnion sheath endothelial linings in the umbilical sheath of amnion vessels, uterus decidua, human fetus epithelium, and non-pregnant human endometrium [115]. Such observations show that regional thyroid status modulation is essential in all reproductive stages in women [115]. Other selenoproteins that are dependent on thyroid status are thioredoxin (TRX) and TXNRD which are redox proteins that have precise sites and positions in placenta of humans and rodents. Analysis of these two proteins indicate that they are abundant in trophoblast, endometrium epithelial, and stromal cells in the stem villi. In addition, they play an important role in guarding placental tissues under inflammatory conditions [16,116,117].

Lower Se levels are reported during different stages of pregnancy suggesting that a stage-dependent relationship is operational. For example, Se deficiency reduces selenoproteins levels involved in redox regulation, impairs placental function and fetal development leading to miscarriage or complicated preterm birth [118,119]. In a clinical trial Se supplementation decreased the risk of pregnancy complications in women at risk of intrauterine growth restriction [120]. However, the fundamental question, which form of Se and how much is beneficial remains unanswered, and this is particularly relevant as Se intoxication in pregnant women was also reported. Thus, controversies exist in routine prescription of Se supplementation to pregnant women and use of Se under these conditions require expert evaluation.

### Selenium Supplementation and Female Reproduction

Sodium selenite supplementation (5 ng/mL) promotes oocyte growth and increases the rate of proliferation in theca and granulosa cells [110]. This impact can, at least partially, be adjudicated through the repression of formation of a free radical—nitric oxide (NO) which causes DNA damage [110]. In another study, the supplementation of the same form of Se but at different concentration (sodium selenite 2.5 and 25 ng/mL, respectively) to the culture medium in porcine parthenogenetic embryos increased the rates of blastocyst formation, the number of cells and inner cell mass rates [121]. Se supplementation throughout pregnancy not only boost antioxidant activity and stimulate production of oestradiol, progesterone and T4, but also enhances overall nutrient metabolism [122].

Recently, Se supplementation (both organic and inorganic forms) has significantly reduced the apoptotic rate in developing follicles in aging mice and improved the rate of in vitro produced pre-implantation embryos [123]. Taken together, although there is no consensus which dose and form of Se is better, a positive impact of Se supplementation is reported on reproduction and pregnancy (Figure 3).

## 5. Selenium and Male Reproduction

The role of Se in male reproduction is relatively well-established and reported compared to female reproduction. Among all selenoproteins, the glutathione peroxidase (GPX) family plays a critical role in many redox responses involved in male reproduction. The key function of these enzymatic isoforms is to defend and guard the cells by catalyzing the reduction of organic hydroperoxides (via glutathione), hydrogen peroxide (H_2_O_2_), and lipid peroxides from oxidative stress as described above. Many of these are tissue-specific and seems to be expressed in sex-specific manners [124] as discussed below.

### 5.1. Glutathione Peroxidase 4 (GPX4)

Additionally, phospholipid hydroperoxide GSHPx, also known as PHGPx, is highly expressed in testes with antioxidant and structural functions. The structural role is prominent because it accounts for over half of the mitochondrial capsule in the mid-piece of a mature sperm (as an oxidatively inactivated protein) and thus plays a critical role in sperm motion [108]. For natural conception, motility is one of the key fertility parameters essential to reach the female oocyte for fertilization. GPX4 is thought to shield the sperm from DNA damage caused by oxidative stress in the initial phases of spermatogenesis, in spite of that, in later stages of spermatogenesis it provides sperm mid-piece stability by being a part of the structure of flagellum-surrounding mitochondrial sheath, an important element in providing integrity and motility to sperm [125,126,127,128].

The significance of GPX4 is illustrated by the death in GPX4 homozygous embryonic mice (KO) after a specific target disruption to GPX4. However, heterozygote mice in gestational stage were unremarkably alive and were born as healthy looking fertile pups [129]. This indicates that not only GPX4 is an essential requirement in the early embryo development but also in gametogenesis. For example, Parillo et al., [130] documented strong expression of GPX4 protein in seminiferous tubules of Chianina bulls at different spermatogenesis stages i.e., spermatogonia, round spermatids, elongated spermatids and cytoplasmic content of maturing sperm. This robust expression of GPX4, while sperm mature at various phases, is consistent with previous physiological findings that GPX4 is crucial for the optimal production and functioning of bovine sperm similar to other mammalian species [130]. Interestingly, the expression of GPX4 in the acrosomal region has been thought to play anchoring role with zona pellucida, highlighting its significance not only at developmental stages of sperm but during fertilization process [130].

### 5.2. Selenoprotein P

Similar with the case with GPX4, selenoprotein P (SelenoP) encoded by the SELENOP gene is thought to have pivotal role in the male reproductive functions. It acts as a Se transporting protein and is also present in vesicle structures in the basal region of Sertoli cells [131]. Sertoli cells are essential in the spermatogenesis process, as not only do they provide anchoring function but also nutrition and endocrine regulation of spermatogenesis in the developing spermatozoa [132]. Male mice with Sepp^(–/–)^ KO (gene responsible for SelenoP protein production) showed considerable reduction in fertility, lower Se levels, and decreased GPX activity [133]. These changes suggest that optimal levels of SelenoP are paramount in sperm functions and fertility. Even high Se diet failed to restore the levels of the testicular Se or the normal phenotype of sperm in Sepp^(–/–)^ KO male mice, which again iterates the significance of Se in spermatogenesis [131].

### 5.3. GPX1 and GPX3

GPX1 and GPX3 selenoproteins, though less abundant are also present in male reproductive tissues and secretions involved in male reproduction. These are well-expressed and defined in sperm and epididymal epithelial cells [134]. In some studies, the epididymal parenchyma and maturing sperm are guarded by these two selenoproteins against the oxidative stress [134]. In addition, GPX3 moves throughout the epididymis lumen with spermatozoa to defend against ROS throughout the maturation process [134].

### 5.4. Selenium Supplementation and Male Fertility

Sperm maturation is closely associated with sperm quality and ejaculation, as well as reproductive efficacy in males. Any anomalies in these processes may therefore lead to insufficient, lower quality ejaculates, and decreased male fertility [135]. Increased consumption of Se-enriched food has been shown to improve GPX antioxidants activity, consequently enhancing fertility in males [136]. For example, Asri-Rezaei et al. [137] reported the Se concentration in testis in male Mus Musculus mice was significantly increased after intraperitoneal injections of sodium selenite (0.50 mg/kg body weight) and Se-nanoparticles (0.50 mg/kg body weight) daily for 7 days (observation made after the 28th day post-injection). Likewise, the antioxidant biomarker enzyme activities catalase (CAT) and glutathione peroxidase (GPX) also increased substantially while decreasing the level of the oxidation biomarker malondialdehyde (MDA) after Se supplementation.

Most importantly, the parameters of the quality of sperm including gross count and motility in the treatment group was also improved compared to control group [137]. These findings generally confirm that Se supplementation can have positive impacts on male reproductive health through reduction of oxidative stress. However, currently not a single form of Se supplementation has added value over others and the doses of available Se supplementations beneficial to male reproductive health are highly debatable. Increased intake of Se supplementations may not offer therapeutic benefit and can even reduce the general reproductive potential in males. While increased production of reactive oxygen species (ROS) leads to oxidative stress, sperm DNA damage and/or apoptosis, membrane peroxidation and decreased sperm motility; an optimal degree of ROS is required to support some essential sperm functions such as capacitation and acrosome reactions [138].

Increased ROS levels have been strongly associated with infertility [139]. Thus, for the optimal functioning of sperm, establishing an equilibrium of Se in the body is important in redox control of male reproduction. Changes in Se level have been shown to affect redox status and induce oxidative stress that might negatively affect male fertility by modifying the expression of relevant biological markers and the behavior of antioxidant enzymes. For instance, Kaushal and Bansal [140] reported that changes in Se levels contributed to oxidative stress through modification of HSP70 protein (heat shock proteins) expressions. Interestingly, a substantial increase in the levels of oxidative stress related markers, including lipid hydroperoxide (LPO), malondialdehyde (MDA), and ROS, was observed in mice with both Se-deficient (0.02 ppm inorganic Se) and Se-excess diets (1.0 ppm inorganic Se).

This outcome demonstrates the need to reconsider which Se form should be promoted for supplementation to the public and how much is needed to maintain homeostasis. In an Se-deficient diet, a reduced GPX level has also been observed, whereas Se-excess group showed comparatively high GPX levels [141]. It is noteworthy that in both categories, the relative markers associated with fertility have decreased significantly [140]. In another related study, elevated LPO, raised oxidative stress and diminished GPX levels were reported in male mice fed with Se-deficient diets [142].

Furthermore, the health benefits offered by organic vs inorganic Se also differ making decision on Se supplementation more complex. For example, a substantial improvement in the overall morphological and histomorphological indexes was observed in testes of young male goats fed with organic Se (Se yeast) of 0.3 mg/kg body weight for 2 months, in addition to the improved GPX enzymatic activity and superoxide dismutase (SOD) in their serum and testicular tissue compared to controls [143]. Another study evaluated the impact of organic and inorganic Se supplementation on ram semen parameters supplemented with either 1.83 mg organic Se (SeMet) per animal per day and 4.0 mg inorganic Se (sodium selenite) per animal per day. After 45 days of Se supplementation, the semen ejaculate volume, sperm motility and viability were significantly higher in Se supplemented diet compared to the control group; organic Se had a much faster effect compared to inorganic Se [144]. In addition, a recent study reported that boars fed on organic Se-based diet (0.5 mg per kg) determined a 23% increase in the sperm concentration per week compared to inorganic Se-based diet (0.5 mg per kg) [145]. Thus, clearly there is marked improvement in the structural and functional factors that define male fertility and sperm functions (refer to Figure 3), but questions on the precise dose, optimal form of Se, and duration of safe supplementation remain unanswered and warrants further research.

## 6. Conclusions and Future Directions

Selenium deficiency leads to several thyroid gland disorders which can impact either synthesis of thyroid hormones or their function at the target tissues which can have implication on global body metabolism. Se supplementation can significantly improve sperm fertility parameters. Although, data on Se and female reproduction are scarce but do suggest that Se supplementation is beneficial in restoring ovarian function and minimizing pregnancy related complications.

Almost all the available literature evidence is based on single time point investigations on either thyroid or reproductive organs studied in isolation. However, given the strong link between thyroid function and reproduction, the root cause must be addressed as targeting reproduction alone will not resolve the issue unless thyroid function is restored that will subsequently improve the downstream associated reproductive functions.

Although Se supplementation has shown convincing results in improving thyroid and reproductive functions, the border between selection of dose and form of Se is very thin as slightly higher doses may lead to Se intoxication. Hence, further well-designed longitudinal studies are required to investigate a range of Se doses and forms to characterize and identify the best form that offers maximum and safe protection.

## Figures and Tables

**Figure 2 nutrients-13-03256-f002:**
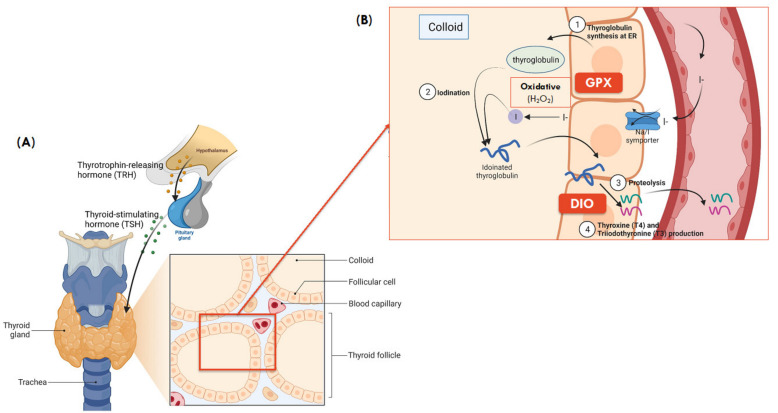
(**A**) Gross and microanatomy of thyroid gland. (**B**) Synthesis of thyroid hormones and role of selenoproteins in colloid of thyroid ( Adapted from “Thyroid Gland Anatomy and Histology”, by BioRender.com (2021). Retrieved from https://app.biorender.com/biorender-templates, accessed on 13 April 2021) and is adapted from [71]. (**1**) Thyroglobulin synthesis: thyroglobulin, the building block of thyroid hormone is synthesized in the endoplasmic reticulum of follicular cells; (**2**) Iodination: iodine then combines with thyroglobulin, forming iodinated thyroglobulin after a series of iodination processes; (**3**) Proteolysis: the iodinated thyroglobulin is broken down into T4 (~90%) and T3 (~10%) hormones; (**4**) T3 and T4 production: Both T3 and T4 then travel in blood and reach different tissues, where T4 is further broken down to the active T3 in local tissues, primarily in the liver to regulate fat and carbohydrate metabolism, thermogenesis, growth, and development [72].

**Figure 3 nutrients-13-03256-f003:**
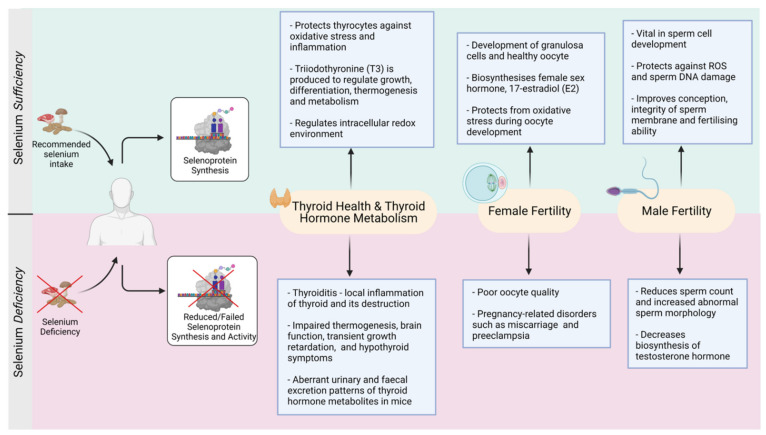
Selenium-mediated balance and imbalance in the thyroid and reproductive functions (Created with BioRender.com, accessed on 2021).

**Table 1 nutrients-13-03256-t001:** Main selenoproteins associated with thyroid hormones functions, antioxidant defense, cellular distribution, and associated pathologies in case of Se deficiency.

Selenoproteins Relevant to Thyroid Health		Cellular Distribution Within Thyroid	Functions Under Optimal Conditions	Thyroid and Other Pathologies Associated with the Respective Selenoprotein Deficiency/Abnormality
Iodothyronine deiodinase (DIO)	DIO1	Plasma membrane	Converts inactive thyroxine (T4) to the biologically active triiodothyronine (T3) thyroid form, and control circulating T3 levels	DIO1 knockout mice had normal growth, development, and fertility, however they exhibited aberrant circulating TH and excretory iodothyronines [53,58]. Se deficiency in chickens inhibited the levels of DIO1, DIO2 and DIO3, which indirectly suppressed the conversion of T4 to T3 [59].
	DIO2	ER membrane	Intracellular conversion of T4 to T3 in thyrocytes, contributing to the major source of T3 in circulation in euthyroidism	DIO2 knockout mice exhibited impaired thermogenesis, brain function, transient growth retardation, in addition to aberrant circulating TH [53,60].
	DIO3	Plasma membrane	Inactivates thyroid hormones from T4 to reverse T3 (rT3), and T3 to T2	Fetal DIO3 knockout in mice is significant among all DIOs. It was characterised by reduced viability, growth retardation, impaired fertility, and hypothyroid symptoms, along with reduced T3 and increased T4 levels in the circulation [53]
Glutathione peroxidase (GPX)	GPX1	Cytosol	Protection against oxidative stress through reduction of H_2_O_2_ and lipid hydroperoxides	Reduced Se-dependent GPX activity, leading to diffusion of H_2_O_2_ into the thyroid parenchyma, followed by local inflammation and ultimately its destruction [61,62]
	GPX2	Not expressed in thyroid	Expression is specific to epithelial cells including colonic crypts and that in gastrointestinal tract and lungs, which offers protection against ingested lipid hydroperoxides [63]	N/A to thyroid health. However, GPX2 was suggested to be the most important Se-dependent antioxidant in colon and early defense against colon cancer [63]. Although no negative impacts of Se deficiency have been illustrated in humans, GPX2 knockout mice when deprived in Se showed enhanced ileocolitis (inflammation) in the intestinal mucosa than that of Se sufficiency [64].
	GPX3	Thyroid colloid (apical side of thyrocyte membrane)	Antioxidant in extracellular fluid; thyroid protection from excessive H_2_O_2_ in thyrocytes and follicular lumen, which is not used by thyroid peroxidase (TPO) for the iodination of thyroglobulin during the process of thyroid hormone synthesis	Se deficiencies reduced Se-dependent GPX activity, leading to diffusion of H_2_O_2_ into the thyroid parenchyma, followed by local inflammation and ultimately its destruction [61,62] In a Taiwanese population, SNPs in GPX3 were significantly associated with the risk of differentiated thyroid cancer [65]
	GPX4	Mitochondria	Reduction of phospholipid hydroperoxides in mitochondria, offering protection of mitochondrial membrane in thyrocytes against peroxidation [61,66] Modulate apoptosis	Limited information on the effects of GPX abnormalities on thyroid function. In Se-deficient mice, GPX4 activity was not significantly reduced when compared to that of GPX1 that had a 50% reduction [67]. Interestingly, embryonic mice with a full GPX4-gene deletion was fatal [68].
Thioredoxin reductase (TXNRD)	TXNRD1	Cytosol	Antioxidant by catalysing NADPH-dependent reduction of thioredoxin, regulating the intracellular redox environment	Limited data on the biological effects of TXNRD abnormalities on thyroid function. However, Se deficiency in mice caused a 4.5% and 11% reduction in TXNRD activity in liver and kidney respectively, while TXNRD level was unchanged [69]. It was postulated that the lack of Sec interferes with the biosynthesis of TXNRD and incorporation into the polypeptide chain at termination, which ultimately produces a non-functional protein [70].
	TXNRD2	Mitochondria	Regulate cell proliferation	

## Data Availability

Not applicable.

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
