# Peer review of "Role for Selenium in Metabolic Homeostasis and Human Reproduction"

_nutrients, 2021, doi:10.3390/nu13093256_

Round 1

Reviewer 1 Report

  1. The review is quite long and really the most essential informations are offered from the Chapter 3 on. Therefore, the final version of the manuscript might benefit from shortening Chapter 1 and 2. In Chapter 2 a lot of information could be included in figure legend. Thus, the main body of Chapter 2 can be shorter, yet comprehensive with no information lost from its current version.
  2. Chapter 6. Conclusions and Future Directions; the text is long and repetitive. It would help the reader to formulate sharp "take-away" messages. 

    I would like to thank the authors for the effort they invested into preparation of this manuscript. 

Author Response

Reviewer 1

The review is quite long and really the most essential informations are offered from the Chapter 3 on. Therefore, the final version of the manuscript might benefit from shortening Chapter 1 and 2. In Chapter 2 a lot of information could be included in figure legend. Thus, the main body of Chapter 2 can be shorter, yet comprehensive with no information lost from its current version.

Thank you very much for your time in reviewing the review and suggestions. As suggested, we have revised the chapter 2 by moving essential text in the figure legend.  We have reduced the text by ~27% (original word count 522 down to 380). Overall, we have curtailed the manuscript by ~ 300 words.

Chapter 6. Conclusions and Future Directions; the text is long and repetitive. It would help the reader to formulate sharp "take-away" messages. 

As suggested by the reviewer we have revised the chapter 6 in single section and have shortened the text by nearly 45% (originally submitted word count 347 down to 190) separated by mini paragraphs each summarising a key message. 

Reviewer 2 Report

This is an excellent review. I have only a few changes to recommend, mostly corrections of minor typographical  errors.

Throughout the document, 'selenium' and 'Se' are used interchangeably. I suggest that after the abbreviation Se is defined, only Se is used, except when the element name is the first word in a sentence, in which case it should be written out in full as 'Selenium'. This would make the review read more smoothly.

Line 11. I suggest changing 'recommended range' to 'the recommended range'

Line 57. Change 'types of selenium has' to 'types of selenium have'

Line 83. Consider hyphenating 'run off' i.e. 'run-off'

Regarding Section 1.4.1., may I suggest that the variation in the Se content of wheat is clarified as being due to the soil in which the wheat is grown. (As an example, bread in New Zealand is made with Australian wheat which contains a good level of Se, because although wheat can be grown in New Zealand, it lacks Se due to the lack of Se in the soil in which it is grown. )

Line 109. Change 'prrimary form' to 'primary forms'

Line 145. Change 'than that' to 'than those'

Line 172. Is it correct to capitalize the first letter of 'aminoacylation'?

Line 221. I suggest changing 'systems thereby,' to 'systems, thereby'

Line 268. Change 'selenoenzymes has' to 'selenoenzym

Line 282. There is a change of tenses here. I suggest changing 'highlighted' to 'highlights'

Line 298. Change 'is only synthesised' to 'are only synthesised'

Line 440. Correct the spelling of 'mesometrial'

Line 496. Change 'thus play'  to 'and thus plays a'

Line 503. The word 'inevitability' does not seem to be the correct word here.

Line 511. I recommend replacing 'comply' with 'is consistent with'

Line 542. Please clarify what species Azri-Rezaei et al did their study on.

Line 552. Change 'healthy' to 'health'

Line 578. Change 'differs' to 'differ'

ine 591. Correct the spelling of  'concentration'

Line 603. Remove the comma after 'Though'

Line 611. I recommend changing 'metabolising' to metabolism'. Also remove the comma after 'though'

Line 624.  Change  'needs' to 'need'

Author Response

Reviewer 2

This is an excellent review. I have only a few changes to recommend, mostly corrections of minor typographical errors.

  1. Throughout the document, 'selenium' and 'Se' are used interchangeably. I suggest that after the abbreviation Se is defined, only Se is used, except when the element name is the first word in a sentence, in which case it should be written out in full as 'Selenium'. This would make the review read more smoothly.

We really appreciate reviewer’s suggestions and time taken to review the document. We have addressed all the typo/grammatical errors in the revised version (highlighted). We have also suggested text in section 1.4.1

Line 11. I suggest changing 'recommended range' to 'the recommended range'

Corrected line 11.

Line 57. Change 'types of selenium has' to 'types of selenium have'

Corrected line 58 of revised manuscript

Line 83. Consider hyphenating 'run off' i.e. 'run-off'

Corrected line 82

Regarding Section 1.4.1., may I suggest that the variation in the Se content of wheat is clarified as being due to the soil in which the wheat is grown. (As an example, bread in New Zealand is made with Australian wheat which contains a good level of Se, because although wheat can be grown in New Zealand, it lacks Se due to the lack of Se in the soil in which it is grown. )

We thank the reviewer for this very relevant example to help the reader understand about the variation. We have added the suggested text in the revised manuscript (line94-98).

Line 109. Change 'prrimary form' to 'primary forms'

Corrected line 110.

Line 145. Change 'than that' to 'than those'

Corrected line 145

Line 172. Is it correct to capitalize the first letter of 'aminoacylation'?

Corrected line 174

Line 221. I suggest changing 'systems thereby,' to 'systems, thereby'

Corrected line 211

Line 268. Change 'selenoenzymes has' to 'selenoenzym

Corrected line 258

Line 282. There is a change of tenses here. I suggest changing 'highlighted' to 'highlights'

Corrected line 272

Line 298. Change 'is only synthesised' to 'are only synthesised'

Corrected line 288

Line 440. Correct the spelling of 'mesometrial'

Corrected line 425

Line 496. Change 'thus play'  to 'and thus plays a'

Corrected 481

Line 503. The word 'inevitability' does not seem to be the correct word here.

Corrected line 488

Line 511. I recommend replacing 'comply' with 'is consistent with'

Corrected line 496

Line 542. Please clarify what species Azri-Rezaei et al did their study on.

Correct name of specie is added line 526

Line 552. Change 'healthy' to 'health'

Corrected line 535

Line 578. Change 'differs' to 'differ'

Corrected line 561

Line 591. Correct the spelling of  'concentration'

Corrected line 573

Line 603. Remove the comma after 'Though'

Addressed and revised the text in conclusions lines 582-597

Line 611. I recommend changing 'metabolising' to metabolism'. Also remove the comma after 'though'

Addressed and revised the text in conclusions lines 582-597

Line 624.  Change  'needs' to 'need'

Addressed and revised the text in conclusions lines 582-597